# Intraoperative Optical Monitoring of Spinal Cord Hemodynamics Using Multiwavelength Imaging System

**DOI:** 10.3390/s22103840

**Published:** 2022-05-19

**Authors:** Nicolas Mainard, Olivier Tsiakaka, Songlin Li, Julien Denoulet, Karim Messaoudene, Raphael Vialle, Sylvain Feruglio

**Affiliations:** 1Department of Pediatric Surgery, Jeanne-de-Flandre Hospital, CHU Lille, Avenue Eugène-Avinée, 59000 Lille, France; 2Laboratoire D’Informatique de Paris 6 (LIP6), CNRS UMR7606, Sorbonne Université, 4 Place Jussieu, CEDEX 05, 75252 Paris, France; songlin.li@lip6.fr (S.L.); julien.denoulet@lip6.fr (J.D.); karim.messaoudene@lip6.fr (K.M.); sylvain.feruglio@lip6.fr (S.F.); 3CERVO, Biomedical Microsystems Laboratory, Université Laval, Quebec, QC G1V 0A6, Canada; olivier.tsiakaka.1@ulaval.ca; 4Clinical Research Group “RIC” Robotics and Surgical Innovations, GRC-33 Sorbonne University, 26 Avenue du Dr. Arnold Netter, 75012 Paris, France; raphael.vialle@aphp.fr

**Keywords:** spinal cord, monitoring, sensors, NIRS, PPG, multispectral

## Abstract

The spinal cord is a major structure of the central nervous system allowing, among other things, the transmission of afferent sensory and efferent motor information. During spinal surgery, such as scoliosis correction, this structure can be damaged, resulting in major neurological damage to the patient. To date, there is no direct way to monitor the oxygenation of the spinal cord intraoperatively to reflect its vitality. This is essential information that would allow surgeons to adapt their procedure in case of ischemic suffering of the spinal cord. We report the development of a specific device to monitor the functional status of biological tissues with high resolution. The device, operating with multiple wavelengths, uses Near-InfraRed Spectroscopy (NIRS) in combination with other additional sensors, including ElectroNeuroGraphy (ENG). In this paper, we focused primarily on aspects of the PhotoPlethysmoGram (PPG), emanating from four different light sources to show in real time and record biological signals from the spinal cord in transmission and reflection modes. This multispectral system was successfully tested in in vivo experiments on the spinal cord of a pig for specific medical applications.

## 1. Introduction

The spinal cord (SC) is a major structure of the central nervous system allowing, among others, the transmission of afferent sensory and efferent motor neurological information [1]. The management of spine and spinal cord disorders frequently requires surgical procedures. In these cases, the placement of metal implants as close to the SC as possible, the correction of spinal anatomy, or direct surgery of the SC involve a significant risk. During surgery, the SC structure can be damaged, resulting in major neurological damage to the patient ranging from transient deficit of superficial sensation to complete paraplegia [2]. During surgical management of adolescent idiopathic scoliosis, the risk of neurological damage can range from 0.5 to over 1% of patients depending on the surgical approach [3,4].

To date, there is no direct way to monitor spinal cord oxygenation intraoperatively as a reflection of its integrity. Indeed, when there is a medullary vascular suffering, the oxygen level is reduced locally. This is critical information that would allow surgeons to adapt their procedure in the event of ischemic spinal cord suffering. It would be particularly interesting to have low cost, intelligent, on-board medical devices, using electronics and optics, during and after the surgical procedure.

Continuous oxygen saturation monitoring using Near-InfraRed Spectroscopy (NIRS) is becoming a classic and widespread tool to monitor tissue vitality and blood perfusion of an organ [5,6,7,8]. It is an optical technology for the non-invasive detection of hemodynamic changes, with advantages of being real-time, continuous, low cost, and portable. Optical techniques for clinical medical applications were developed after the invention of muscle oximeters by G.A. Millikan in the 1940s [9]. In 1977, the first in vivo application of NIRS was described by F.F. Jobsis [10]. He found that the relatively high transparency of biological material in the NIR region (650–1000 nm) allowed the non-invasive measurement of tissue oxygen saturation (SO_2_) in real time using the principles of optical spectrophotometry. In 1993, NIRS was approved by the U.S. Food and Drug Administration (FDA) for non-invasive, continuous monitoring of cerebral and somatic tissue oxygenation [11,12]. It is a technique that uses a laser diode and/or light-emitting diode (LED) light sources covering the optical window between 650 and 1000 nm, and flexible optical fibers to transport NIR light to a source and from a detector [13,14]. NIRS is based on the fact that: human tissues are relatively transparent to light in the NIR spectral window; NIR light is either absorbed by pigmented compounds (chromophores) or scattered in tissues; NIR light is able to penetrate human tissues since the dominant factor in its tissue transport is scattering [15]; the relatively high attenuation of NIR light in tissues is due to the main chromophore hemoglobin (the oxygen-carrying protein of red blood cells) located in the small vessels (<1 mm in diameter) of the microcirculation. NIRS can differentiate the absorption spectra of most important blood chromophores: oxygenated hemoglobin (HbO_2_), deoxygenated hemoglobin (HHb), methemoglobin (MetHb) and cytochrome aa3 (CCO). Hence, it can measure relative changes in oxygen concentration using these four chromophores [7].

The aim of this study is to develop and test in vivo a multispectral imaging sensing system to monitor in real time the oxygenation of the SC during spinal surgery, based on NIRS in combination with other additional sensors, including ElectroNeuroGraphy (ENG). The device, working with multiwavelength, simultaneously uses both reflection and transmission modes for Near-InfraRed Spectroscopy.

The additional parts of the article are composed as follows. Following the introduction, our hardware and software developments are presented in Materials and Methods. Then, the experiment on a large animal (pig) and its results are presented starting with the raw results then their analysis in reflection and transmission to finish with the extraction of physiological parameters from these values. In the discussion part, after a brief reminder of the results and thanks to a review of the complete literature, we review the interest of developing such a device, the importance of monitoring several chromophores in the blood and we discuss the use of the transmission mode rather than the reflection mode. Finally, we situate our experimentation in relation to the existing studies.

## 2. Materials and Methods

### 2.1. Acquisition Platform Design Overview

The proposed instrument was designed for the continuous wave monitoring of living tissues, using functional NIRS (fNIRS) technique with four wavelengths in correlation with other sensors in the region of interest. Figure 1 shows the block diagram of the system implemented for our study, which is composed an acquisition board and probes. The whole system is battery powered to reduce mains-related noise and parasitic signals. Two supply voltages are then generated thanks to Low DropOut (LDO) regulators.

#### 2.1.1. Acquisition Board

The main board is structured around an MBED NXP^®^ LPC1768 microcontroller (MCU) with an ARM Cortex-M3 (32-bit, 96 MHz). It was chosen for its versatility, flexibility and software development simplicity. It is connected via USB to a computer acting as base station to allow dynamic reconfiguration and data recovery. In the context of this exploratory SC study, a very high-performance Analog to Digital Converter (ADC) had to be chosen. We settled on the ADS1298 24-bit delta-sigma (ΔΣ) ADC by Texas Instruments^®^, which presented the best trade-off. It includes eight independent channels (i.e., simultaneous sampling), functioning at high sampling resolution (maximum of 32 kilo-Samples per second, kSps) and featuring an integrated Programmable Gain Amplifier (PGA). Two channels were dedicated to NIRS to obtain optico-electrical signals, known as PhotoPlethysmoGrams (PPGs). T-probes were used for transmission mode and R-probes for reflection mode. In addition, four channels (for four electrodes) were reserved for two differential biopotential measurements (ENG). This ADC is linked to the MCU by a Serial Peripheral Interface protocol. Moreover, analog Low-Pass Filters (LPF) were implemented between probes and ADC to minimize the intrinsic noise bandwidth and parasitic signals. The cut-off frequency was fixed to 153 kHz for antialiasing close to the ΔΣ modulator sampling frequency.

An SMT160-30 digital output temperature sensor (Smartec^®^, Breda, The Netherlands) is included and is directly connected to the MCU. A synchronized digital input to follow stimulation completes the system. The LEDs can be commanded by a driver module running on the MCU, which is connected to the common anode of the light sources. However, additional Bipolar Junction Transistor (BJT) circuit is also present to drive higher LED current and, therefore, to have a more important optical intensity. A photograph of the acquisition board, made from an FR4 Printed Circuit Board (PCB), can be seen in Figure 2.

#### 2.1.2. Trans-Reflectance Optodes

Our system is designed to operate simultaneously in transmission and in reflection (Figure 3). The T-probe in Figure 1 was based on TAOS^®^ TSL12S and the R-probe on TSL12T. They feature the same high-speed photodetector (with a rise time of a few µs) that includes an integrated TransImpedance Amplifier (TIA) with a gain of 80 MΩ in the range of 320 nm to 1050 nm. They generate an analog voltage output between 0 and 5 V, perfectly suited to the selected ADC. Note that the analog front-ends currently available on the market for this specific application are not suitable because of their limited resolution. Indeed, the compromise between amplitude and time resolution is not admissible here. These sensors are mounted on two different packages. The TSL12T chip, more compact than the other, is more suitable for our PCB design. Moreover, the TSL12S has an improved response of 2.1 (experimental measurement) in comparison to TSL12T, thanks to its optical lens (with identical photosensitive surfaces of 0.05 mm × 0.05 mm).

In order to achieve a perfect alignment of light sources in R-probe for the T-probe, the two probes were fixed on each side of an adjustable spacer clamp (Figure 4).

For the R-probe, we designed a custom FR4 PCB in order to combine the light sources and the optical sensor. The probe dimensions (14.2 mm × 14.2 mm) are compatible with the primary application [16]. As can be seen in Figure 5A, two dual wavelength LEDs are placed on both sides of the TSL12T. The light sources are equidistant from the photodetector, allowing a cross-comparison of the extracted parameters from the two pairs. This probe is positioned directly in contact with the spinal cord. Because of the need to adapt the probe design to the local anatomical constraints, we had to choose a short distance between the photosensor and the LEDs (4.32 mm, see Figure 5B), and as a trade-off in order to obtain the best response for the four wavelengths. This short distance may result in a low diffusion depth that must be taken into account when interpreting the results. Liquide epoxy was applied to the whole surface of the PCB tracks and pads to waterproof the probe during in vivo animal experiments, leaving the optical windows of the optoelectrical components untouched. Moreover, a series of optical barriers of different heights were developed to reduce self-glare and direct-light. Figure 5C shows the 3 mm high walls used during the experiments. 

In accordance with the chromophores that we aimed to monitor [17,18], we chose the Marubeni^®^ SMT660/890 and SMT735/850 LEDs. Note that green LED is not suitable due to its low penetration depth in tissue. The LEDs’ optical characteristics had been measured previously, such as the peak wavelength and the Full Width at Half Maximum (FWHM), and they showed good compatibility with the photodiode′s spectral responsivity (Figure 6). The protocol of the optical characterization is described in a previous study [19].

#### 2.1.3. Signal Acquisition and Pre-Processing 

All probes connected to the ADC are converted at 4 kSps. The MCU acquires the digitized signals and converts the 24-bit values into their hexadecimal equivalent char to reduce the amount of data to be transmitted. The resulting word is bundled with the corresponding indicator of the illuminated light source at the time of the conversion before sending it to the computer via the serial link. The data rate is adapted to allow a minimum delay.

The initialization sequence is set from the computer, where a custom Python process launches the acquisition procedure on the MCU, creates a timestamped datalog file and listens on the serial port for new data. Upon the incoming of a packet, the data is converted back to its numerical value and stored into the log file, where the data vectors are stored according to their origin. In parallel, a Kst-plot interface, KDE, is set to display in real time all of the acquired raw data. Note that the four LEDs are successively activated by the MCU. They are illuminated at a period of 4 ms and operate in pulsed mode with a reduced high state (10 µs) in order to minimize consumption and heating. Thus, we have 1 kSps for each wavelength on both photodetectors with an additional measurement of darkness, when no LED is active.

After reception, offline analysis of signals was performed with MATLAB, MathWorks, on the base station to improve the data quality and extract relevant information, which are for us ENG, Breath Rate (BR), Heart Rate (HR), the chromophore concentration changes (Δ[X]) and the oxygen level evolution (ΔSpO_2_). In particular, high-order filters were used to improve the Signal-to-Noise Ratio (SNR), while minimizing group delay and with flat response in the passband, which could deform the signals’ shape [20]. As illustrated by Figure 7, from raw signals, ENG is directly observed after High-Pass Filter (HPF). It is a 2000-order Hamming Finite Impulse Response (FIR) with a cut-off frequency of 100 Hz. Additional LPF has no positive effect on parasitic signals and intrinsic noise. Concerning PPG signals, the pre-processing to obtain the BR only requires the removal of the Direct Current (DC) component. This is achieved using a Blackman HPF with a cut-off frequency of 1 Hz. For the HR, a Band-Pass Filter (BPF) was chosen (a 0.5 Hz 2000-order Hamming HPF associated with 5 Hz 2000-order Blackman LPF). Conversely, concerning the obtaining of the concentrations’ variation, it is the rejection of a specific band, with a custom Band Stop Filter (BSF), which is recommended. In particular, the contribution of breathing is now removed without modification of the DC component. After this shaping of the raw signals, it is possible to obtain the physiological parameters.

### 2.2. Physiological Estimations Parameters

To monitor the functional status of the SC, the blood composition in this region of interest is indeed primordial in association with HR and BR. Four different wavelengths were chosen to detect changes in the concentration of four different substances in the SC blood thanks to PPGs. Figure 8 shows the molar extinction coefficients of the main substances of interest in blood in function of the light wavelength based on the literature [21]. We decided to observe the concentration changes of HbO_2_, HHb, MetHb and CCO [22]. The carboxyhemoglobin (COHb) was not selected due to its relatively small value compared to the other substances, and its low concentration in this study.

According to the Modified Beer–Lambert Law (MBLL) [23], the UCL4 algorithm [18,24] was applied to calculate the concentration evolution of these elements through this matrix equation
(1)[Δ[HbO2]Δ[Hb]Δ[MetHb]Δ[CCO]]=1d[εHbO2(λ1)εHb(λ1)εMetHb(λ1)εCCO(λ1)εHbO2(λ2)εHb(λ2)εMetHb(λ2)εCCO(λ2)εHbO2(λ3)εHb(λ3)εMetHb(λ3)εCCO(λ3)εHbO2(λ4)εHb(λ4)εMetHb(λ4)εCCO(λ4)][ΔA(λ1)DPF(λ1)ΔA(λ1)DPF(λ2)ΔA(λ1)DPF(λ3)ΔA(λ1)DPF(λ4)]
where Δ[X] is the concentration change of the chromophore X. d, the physical distance between the light emitter and the photodetector, has a different value in reflection and transmission. For transmission, it is the cross-sectional diameter of the SC, which is 1 cm. For the reflected lights, it is set at 4.32 mm due to our design constraints. ε_X_(λ_i_) corresponds to the molar extinction coefficient at the wavelength λ_i_ (with i = 1, 2, 3 and 4). The values for the selected wavelengths are shown in Table 1. ΔA(λ_i_) is the change in light absorption at the wavelength λ_i_. It is defined as follows
(2)$$\Delta A\left(\lambda_{i}\right)=A\left(\lambda_{i},t_{1}\right)-A\left(\lambda_{i},t_{0}\right)=-\log_{10}\left[\frac{I\left(\lambda_{i},t_{1}\right)}{I\left(\lambda_{i},t_{0}\right)}\right]$$
where I(t_j_, λ_i_) is the light intensity of the wavelength λ_i_ received by the photodetector at the time t_j_ and t_j_–t_j-1_ is the sampling period, fixed at 1 ms. The normalized values for the Differential Pathlength Factor (DPF) for the reflection and transmission lights (Table 2) are estimated thanks to the literature [25,26].

Classically, the pulse oxygen saturation level (SpO_2_) is obtained as the ratio of [HbO_2_] on all hemoglobin concentrations, such as
(3)$${\mathrm{SpO}}_{2}=100\frac{\left[{\mathrm{HbO}}_{2}\right]}{\left[{\mathrm{HbO}}_{2}\right]+\left[\mathrm{Hb}\right]+\left[\mathrm{MetHb}\right]}$$

Knowing the chromophores concentration changes thanks to Equation (1), a recursive method is used to calculate the SpO_2_ variation, as follows
(4)$$\Delta {\mathrm{SpO}}_{2}\left(t_{i}\right)={\mathrm{SpO}}_{2}\left(t_{i}\right)-{\mathrm{SpO}}_{2}\left(t_{i-1}\right)$$

Concerning HR and BR, they are obtained after peaks and valleys detection on PPG signals (periodic minimum and maximum). For BR, sliding window filter with a length n of 2 samples is added to obtain a smooth curve of breathing rate. Then, BR, in breaths per minute (Bpm), is obtained as follows for maximum values
(5)$${\mathrm{BR}}_{\mathrm{peak}}\left(m\right)=\frac{60}{\left\{\frac{t_{\mathrm{BR}}\left[\mathrm{peak}\left(m\right)\right]-t_{\mathrm{BR}}\left[\mathrm{peak}\left(m-n\right)\right]}{n}\right\}}$$
where t_BR_[peak(m)] is the time associated to the m-th peak. Then, HR, in beats per minute (bpm), is deduced with the same method, with n = 7 for better averaging and after deletion the breath in the acquired signals.

## 3. Results

### 3.1. Pig Experimentation

In vivo experiments were performed with veterinarians of the Veterinary School of Alfort (EnvA-Crbm), after obtaining permission from the French ministry for research and ethical evaluation by the animal care and use committee number 16 (EnvA-anses-u-pec), under the authorization number 20287. The pig (Sus scrofa domesticus) was chosen as the experimental animal because it represents the closest animal model to humans in terms of vertebral anatomy and cardiovascular system [27,28].

The animals were put to sleep during the open surgery trials and sacrificed without waking up afterwards. During the whole procedure, the animal was under general anesthesia according to the procedures in force, associating a gas anesthesia with isoflurane (1.5 to 2%) firstly and an analgesia by morphinics (morphine in intravenous continuously), and after with Propofol to prevent unwanted effects of isoflurane in our experiment. During the whole procedure, the main vital parameters (body temperature, heart rate, tele-expiratory oxygen pressure, etc.) were monitored by conventional equipment. The animal was positioned on ventral decubitus for the surgery, exposing the vertebral column from the ninth thoracic vertebra to the fifth lumbar vertebra. A laminectomy was performed on the third lumbar vertebra, which is the removal of the posterior part of the vertebra to expose the SC. The device was then positioned in contact with it (i.e., on the dura matter). In addition, subdermal needle bipolar electrodes to stimulate and to record electrical activity were implanted. Stimulation electrodes were placed near the sciatic nerve, while recording electrodes were in the nerve roots area in the periphery of the third lumbar vertebra (L3), where the optical probe was already placed. The electrostimulation system used was a STIMOLA device, from Biopac^®^. It is controlled using the AcqKnowledge 5.0 software and monitored in parallel with our system by the data acquisition system MP36R of Biopac^®^. The pulse trains were pre-programmed to deliver current pulses of 10 mA lasting 100 µs under 1 V, with rise and fall times of 20 µs and a minimum deviation of 3 ms between two successive pulses.

### 3.2. Raw Signals

During any in vivo experimentation, raw signals must be observed in real time in order to check whether they are in accordance with expectations before any digital signal processing (Figure 9). Note that, in our preliminary work, elementary materials were used and preliminary tests on reference elements were indeed carried out. Optical spectroscopy measurements were been performed during the in vivo test and optical elements were also characterized [16]. We are now working in relative value, thanks to Equation (2) to minimize calibration during in vivo experiments. As can be seen, these raw in vivo signals are heavily noisy, but a trend emerges for PPG signals with a periodic pattern and a slight decrease in the mean value of the signals, typically a sign of oxygen desaturation in this transient phase. Motion artifacts also affect the measurement. In particular, the PPGs’ modulation by the respiratory movement is mostly present in transmission mode. On the reflection mode, there are mainly classical electrical perturbations. Moreover, the reflection measurement suffers from saturation on some value captures (not represented here), mainly due to the surgery process, where blood can enter into contact with the photodetector. The accumulation of blood causes too much reflection, which creates a saturation of the output voltage of the optical sensor. It is not generally the case with the other sensor in transmission.

Regarding the Alternative Current (AC) components of these data, nothing can be said at this stage and signal processing must be carried out. However, it is interesting to observe the DC components. The received lights in reflection mode is on average greater than in transmission (the photodiode output increases with the light intensity). This is in line with our expectations because the light path length between emitter and receiver of the reflective system is shorter and we have, therefore, a lower number of biological layers (as a result of the penetration depth). In both cases, the light sources at 660 nm and 735 nm give the extreme values, with the 735 nm light source presenting the higher DC output response. We can also notice that the 850 nm light source gives a higher response than the 890 nm one in transmission mode, but that it is the opposite in reflection mode. This may be due to the different mediums that the reflected light and the transmitted light have to pass through.

### 3.3. Reflection Mode Versus Transmission Mode

After filtering the signals, we processed the AC characteristics. In Figure 10, we can see eight PPG signals in transient. Unlike DC components, the AC responses in transmission are higher than in reflection. Indeed, we have a factor of 2.5 between both modes (factor 5 divided by the difference in response of the two photosensors, see Section 2.1.2). Thus, the vascular-related markers (systolic and diastolic wave, dichotic notch, etc.) generally used are more visible with the T-probe. Moreover, compared to the other, we note an inversion in the PPG signals in reflection. It should be noted that this was not the case during our tests on the finger (considered as the golden standard). The signals in this case were all in phase. We can assume that this comes from the fact that we do not really monitor the same region and, therefore, we do not observe the same blood flows (at the macroscopic level). Finally, in reflection mode, PPG signals do not always exhibit the same pattern for all wavelengths at the same time, which seems still to be related to the penetration depth. Thus, the observed Blood Flow (BF) path simply differs between R-mode and T-mode, which could be associated to the elastic deformation of tissues, mainly of capillaries (phenomenon related to the pulse wave) [29,30].

Correlation analysis was also performed to compare results of both modes. Firstly, a cross-correlation coefficient of 1 is obtained for zero-time delay, which confirms the strong periodic link of both acquisition methods. Moreover, as we have gaussian distributions, the Bravais–Pearson coefficient between both probes results for all selected wavelengths were calculated. As all coefficients are closed to 0.8 in absolute value (Table 3), the linear relation is validated, and, to go further, we traced on the bottom of Figure 10 four scatter clouds. They present the relation between the four pairs of PPG signals in reflection (*Y* axis) and in transmission (*X* axis). Two regression lines were also defined (R-probe results in term of T-probe results and the opposite). For strong correlation, the angle between the two lines should be as small as possible. With 8 degrees, in the worst case, the strong correlation between both methods is confirmed. However, the information in the transmission part is 2.5 times more important.

### 3.4. Extracted Physiological Parameters

Figure 11 presents the case study where current pulses are applied. We have five different phases (see Figure 11a). At the beginning and end, no intensive stimulation is carried out. Between 26 s and 40 s, a first series of pulses is performed. Then, until about 55 s, the stimulation is slightly less intense and, finally, between 55 s and 70 s, trains of pulses are sent, similar to what happens for the evoked potentials. As shown in Figure 11a,b, the relation between the electrical stimulation of Figure 11a and the optical signals behavior is not clearly visible. However, after the processing presented in Section 2.1.3, relevant information with high added value can be obtained. As an example, Figure 11c represents the calculation of the spontaneous breathing (the plot delay is due to the computation method used). Beginning around 15 Bpm, BR perfectly follows the trend associated with the five stimulation phases.

In Figure 12, the resulting temporal variation in the four main blood chromophores’ concentration (Δ[X]), ΔSpO_2_ and the HR are proposed. We can clearly observe how the HR increases along with electrostimulation. Concerning Δ[X] and ΔSpO_2_, the breathing requires us to smooth the curves in order to be able to deduce a trend. We see the benefit of using more than two wavelengths when we look at the SpO_2_ variations. With only two light sources (raw classical in Figure 12b), no variation is detectable. On the other hand, with the four selected wavelengths (raw multiwavelength), a trend similar to the one observed with HR can be identified.

Concerning the ENG aspect, Figure 13 shows an example of the variation in biopotential at the level of the SC. It was obtained thanks to the MP36R with a resolution of 200 kSps under 24 bits. This curve was obtained after filtering and, above all, an overall average over 15 periods in order to extract signal from noise. We can see a first pulse due to the electrical stimulation and, about 1.9 ms later, the sign of the neurons’ recruitment, with their associated oxygen consumption.

## 4. Discussion

The results of this study show that this multispectral system has successfully allowed in vivo experiments on the spinal cord of a pig simultaneously using multiwavelength devices with high resolution in reflection mode and transmission mode. After real time observation of raw signals to ensure that they are as expected, we were able to extract precise physiological and hemodynamical parameters (e.g., HR, SpO_2_, ENG) thanks to the developed systems.

The gray matter (brain and SC) is the functional center of the nervous system. The basic cells that compose it are the neurons. The neuron is a cell anatomically and physiologically specialized in the reception, integration and transmission of neurological information. Each of them is integrated in multiple, ordered and hierarchical networks responsible for receiving or transmitting a signal or coordinating a complex function. In this context, any damage to the integrity of the SC can lead to an impairment of motor, cognitive and vital functions [31]. When performing spinal surgery, such as scoliosis management, the SC can be damaged and lead to transient or irreversible deficits resulting in major disability for the patient. Therefore, it appears essential to manage patients with acute spinal cord injury (SCI) as best as possible in order to minimize their long-term neurological deficits. Several studies have already shown that during spinal distraction, as well as during surgical treatment of scoliosis, the appearance of vascular disorders temporarily precedes the occurrence of SCI [32,33,34]. Currently, there is no way to monitor the integrity of the SC directly after any type of trauma. This tool, adapted to the anatomical constraints and the intraoperative context, would allow the surgeon to adapt his decision making in real time in order to manage this critical situation in as optimized a way as possible. Available imaging techniques such as magnetic resonance imaging, considered as the reference method to evaluate all lesions associated with spinal trauma [35,36], are not yet available to the surgeon in the operating room. Emerging invasive options offer new modalities for the evaluation and treatment of SCI [37]. For example, the implementation of an epidural or intrathecal pressure monitoring device can provide a better understanding of SC perfusion pressure parameters after various procedures [38,39]. Much of this research is in its beginning stages, but ongoing efforts are improving understanding of the importance of SC perfusion on recovery from SCI.

The technique currently used in the operating room to monitor the state of the SC is based on electrophysiology, with ENG and evoked potential [40,41]. Using electrodes implanted invasively or on the surface of the skin, the sensory or motor electrical signals (bio-potentials) that run through the tissue can be characterized. Indeed, when the nervous system is affected, the sensory or motor potentials collected are altered [42,43]. However, these techniques suffer from an essential inconvenience. The observation of electrophysiological signals is not able to prevent the damage but only to observe it afterwards. These methods do not allow the identification of early metabolic dysfunction, particularly related to vascular phenomena, before neurological functions are affected. The observation of an electrophysiological modification is a sign of an already installed suffering. It is therefore an important but indirect and delayed information that does not always allow to go back and prevent the lesion. In this context, the NIRS technology responds exactly to this problem by providing a real-time monitoring tool capable of monitoring the SC perfusion.

Continuous monitoring of tissues’ oxygen saturation is therefore a standard measurement in daycare. However, several concerns about classical dual wavelength pulse oximeters remain relevant. In order to increase precision, monitoring other blood chromophores, some approaches employed more than two light sources to realize co-oximeters and signal processing [44,45,46]. Those non-conventional multiwavelength probes are designed to address those limitations by quantifying other hemoglobin derivatives and other chromophores and, thus, to improve precision of the tissue perfusion level and its composition through multiple simultaneous PPGs at different depths. Liu et al. [17] realized a clinical comparative study on 20 subjects that showed that the multiwavelength PPG method significantly improves the measurement accuracy of blood pressure, reducing the mean absolute difference between the reference and the estimated systolic blood pressure values from 5.7 mmHg (for single-wavelength PPG) to 4.0 mmHg (for two-wavelength PPG) and 2.9 mmHg (for three-wavelength PPG). The light sources present on our probe were carefully chosen in order to present couples of usable wavelengths for oximetry. Used together, these different couples of sources allow a measurement by Diffuse Optical Imaging (DOI) to be performed on four distinct wavelengths, namely 660 nm, 735 nm, 850 nm and 890 nm. This allowed the comparison of the performances of the measurement according to the couples of selected wavelengths. We note that with this type of prototype, co-oximetry is also feasible and thus allows us to appreciate the fraction of hemoglobin really involved in the transport of oxygen to the tissues, but also to contribute to the diagnosis of pathological states related to the increase in certain fractions (COHb, MetHb, SulfHb). 

Usually, this kind of optoelectrical system is applied only in reflection mode (R-mode) as it is easier to position the device on tissue and its energy consumption is lower than in transmission mode (T-mode) [7,44,47]. In this work, both transmission and reflection modes were utilized to gather the hemodynamics of the SC. Although several studies chose to focus on/explore the reflective part, we note that the features from the transmitted light signal are more relevant regarding the vascular changes in this specific region within the central nervous system. Physiologically, signals measured in transmission mode ensure the majority of the photons go through all the vascular network of the SC section and capture an informative depth-related BF signal from the spine, contrary to the reflection signals. Indeed, the results confirm this observation with a greater pulsatile response. This characteristic gives a higher SNR when considering the AC portion for feature extraction (HR, blood pressure, respiration). Moreover, in our study, special care had to be taken to reduce the autoglare by using 3D printed windows. The problem of self-glare was taken into account in the design of this reflection probe. For this, we used 3D printing technology to manufacture a mask, with custom windows, designed with the FreeCAD^®^ software (Figure 5C). The height of this optical mask is a crucial parameter in the performance of such a device. Several masks were manufactured to be tested on a test bench, namely 3 mm, 4 mm and 5 mm. The selected height was 3 mm. Indeed, the optical effects at the interface of a multilayered medium are to be taken into consideration in the design of a reflective device, which is not needed in transmission. Finally, the transmission mode PPG is less disturbed by volume changes in the media, originated by the BF and the pression on the tissues, which is not an issue here. This comparison must be balanced by the fact that the distance between the emitting and the receiving sources is too small, resulting in a low penetrance in the analyzed tissue. However, this probe was created for a specific application, spinal surgery, and must therefore respect certain imperatives, notably anatomical. This comparison is therefore made in this specific context and cannot be generalized to all NIRS uses.

Hemodynamic monitoring of SC using NIRS is a promising technology that has been in constant development for several years. It follows closely the research on its use in traumatic brain injury for which the literature is already abundant. Indeed, M. Roldán et al. published in 2021 a systematic review of the literature on the use of NIRS in traumatic brain injury and found 72 publications covering the period from 1977 to 2020 that were directly relevant [47]. In this article, all the papers that assessed paired oxygenation measurements reported positive results using NIRS as a diagnostic tool, which can be considered as a significant change in NIRS signals before and after an intervention (change on mean arterial pressure, hyperoxia, hypercapnia, etc.) [48,49,50]. Furthermore, in cases where NIRS-derived oxygenation parameters were compared with variables not directly related to oxygenation, such as intracranial pressure, cerebral BF or mortality, the authors also reported positive results [51,52,53]. However, despite the abundance of literature showing a growing interest in the subject, the authors conclude in this review of the literature that, despite the positive results that NIRS showed in these research studies, there is still more work to be carried out in comprehensively evaluating NIRS in order for it to be established as a reliable and routine monitoring technique in trauma brain injury.

Despite the relative success of NIRS techniques in cerebral monitoring, similar applications for monitoring of the SC have been limited to date. For comparison, a systematic review of the literature published in 2019 found 26 articles focused on direct/indirect monitoring of SC BF and oxygenation on animal (n = 17) or human (n = 9) [7]. In 2002, A. Macnab was the first to demonstrate the feasibility of using NIRS to monitor the SC [54]. They manipulated oxygen saturation and BF in the SC of three pigs and showed immediate corresponding changes in the concentration of oxygenated, deoxygenated and total hemoglobin in the cord. Thus, they concluded that NIRS monitoring could help surgeons prevent cord damage by providing real-time detection of the onset of ischemia and hypoxia, and hence allow intervention capable of restoring perfusion and oxygen delivery so as to prevent irreversible neuronal injury. Subsequently, several more recent studies have been conducted, notably on rabbits [55], sheep [56,57,58] or pig [59,60,61,62].

In 2019, B. Shadgan et al. [60] investigated the feasibility and validity of using a miniaturized multiwavelength NIRS sensor for direct transdural monitoring of spinal cord oxygenation in an animal model of acute SCI. It worked in reflection with a distance between emitters and receiver close to ours. Nine Yorkshire pigs underwent a weight-drop contusion-compression injury and received episodes of ventilatory hypoxia and alterations in mean arterial pressure (MAP). They showed that NIRS parameters of tissue oxygenation were highly correlated with intraparenchymal measures of tissue oxygenation. In particular, during periods of hypoxia and MAP alterations, changes in NIRS-derived SC oxygenated hemoglobin and tissue oxygenation percentage corresponded well with the changes in SC oxygen partial pressures measured by their intraparenchymal sensor. This study confirmed the results of the preliminary studies carried out on a pig model [62,63]. In 2018, D.R. Busch et al. [56] focused on demonstrating the safety profile of the required light exposure for the SC. It was the first report of in situ safety testing of this technology. They exposed the SC of 11 adult sheep to laser light utilizing a custom fiber-optic epidural probe and evaluated the tissue illumination, neurological and pathological outcomes of the irradiated sheep and heating in ex vivo SC samples. They concluded that low tissue irradiance and the lack of neurological, pathological and temperature changes upon prolonged exposure to the laser source offer evidence that SC tissues can be monitored safely with NIR optical probes placed within the epidural space. In 2020, the same research team developed an epidural optical device capable of directly measuring and immediately detecting changes in SC BF using diffuse correlation spectroscopy during spinal distraction [58]. It was continuously monitored at the distraction site in 10 sheep and was significantly lower than baseline in the 10 min after maximal distraction with a BF of less than 40% of the baseline. After release, BF at the distraction site recovered to a median BF of −13% (−31%, −5%). It demonstrated high temporal resolution and the capacity to axially resolve changes in spinal cord BF at and remote from the site of distraction. These early results suggest that this technology may assist in the surgical management of spine trauma and in corrective surgery of the spine. In our experiment we did not study phases of medullary hypoxia or spinal distraction. Therefore, we studied our probe only in lesion-free conditions. In this context, we cannot demonstrate for the moment any correlation between the decrease in SpO_2_ measured with this device and a spinal cord injury.

Regarding human studies, most of them use transcutaneous devices [29,30,64,65,66] based on the “paraspinal collateral network” concept [67]. This states that blood supply to the SC is provided by a rich network of paraspinous arterial collaterals, which are also shared with surrounding tissues including the paraspinous muscles. The results are currently rather heterogeneous, and although there may be a correlation, the authors agree that this presents a significant delay and that further research and experimentation is necessary. The small diameter of the spinal cord and the depth of the overlying tissue have made it difficult to assess SC perfusion in human subjects without a direct surgical approach.

A.R. Amiri et al. in 2013 [68] investigated whether it is possible to monitor physiological changes in human SC perfusion intraoperatively using NIRS with indocyanine green tracer technique. It is a no diffusible, fluorescent, water-soluble tricarbocyanine dye that is a strong infrared absorber [69]. SC perfusion was measured intraoperatively in 18 patients undergoing elective posterior cervical spinal surgery. It was the first study to investigate the use of NIRS during spinal surgery in human subjects. They were able to clearly identify a significant increase in SC perfusion after hypercapnia through transdural and translaminar measurements.

Compared to this literature, our system is therefore modular, flexible, low cost and portable. It allowed us to confront, simultaneously, different approaches with success in a specific and strongly constrained surgical context. However, additional improvements are desirable. First of all, in an intraoperative context, making the system more embedded by a wireless communication to the base station would be judicious. The amplitude resolution is more than satisfactory. The temporal resolution, well adapted for the optoelectronic part, could be increased for the ENG part, in order to approach a measurement bandwidth around 10 kHz, and thus to free oneself from the Biopac^®^ acquisition equipment. The signal processing aspect allows us to obtain state-of-the-art results. It could also be deepened to make the system intelligent and better assist the surgeon. The objective of the next studies to be carried out is to create a probe validated in animals, easily usable and sterilizable by conventional procedures in order to be able to test it on a large number of patients in real conditions, and thus to be able to draw conclusions allowing its use in the everyday life of the spinal surgeon.

## 5. Conclusions

This multispectral system was successfully tested in in vivo experiments on the spinal cord of a pig for specific medical applications. Its main strength is the simultaneous use of multiwavelength in reflection mode and transmission mode with high resolution. Although this device needs to be tested further in vivo to validate the measurements and their reproducibility, it is positioned as a promising tool to monitor the oxygenation of the spinal cord intraoperatively during spinal surgery, reflecting its integrity. Once validated for spinal cord applications, this device could be used for other organs such as the liver or the heart during various invasive procedures.

## Figures and Tables

**Figure 1 sensors-22-03840-f001:**
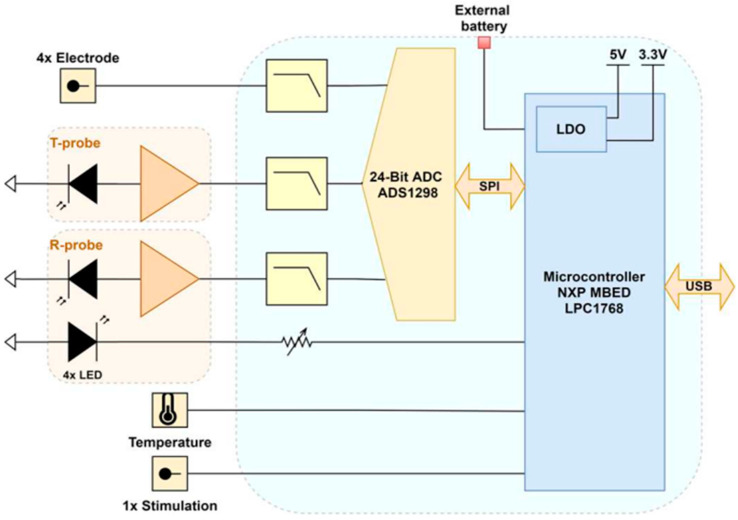
Block diagram of the multimodal device, with various probes on the left and the acquisition board on the right (SPI: Serial Peripheral Interface; ADC: Analog to Digital Convertor; LDO: Low DropOut).

**Figure 2 sensors-22-03840-f002:**
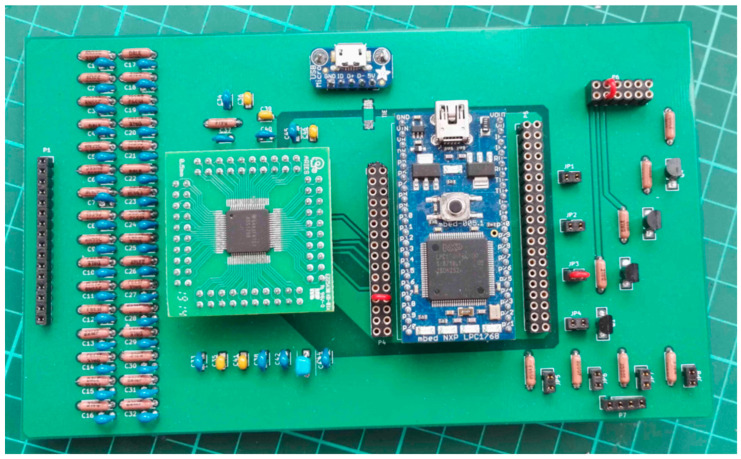
Acquisition board only, where inputs are on the left and outputs for LED are on the bottom right.

**Figure 3 sensors-22-03840-f003:**
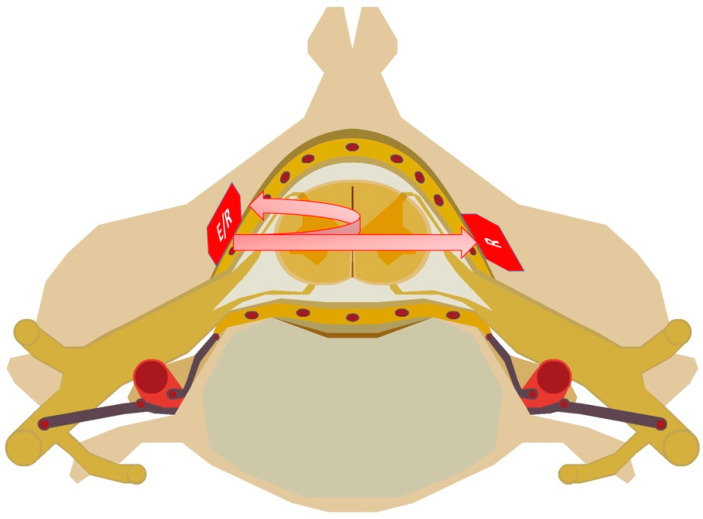
Illustration of the optical imaging process with custom probes.

**Figure 4 sensors-22-03840-f004:**
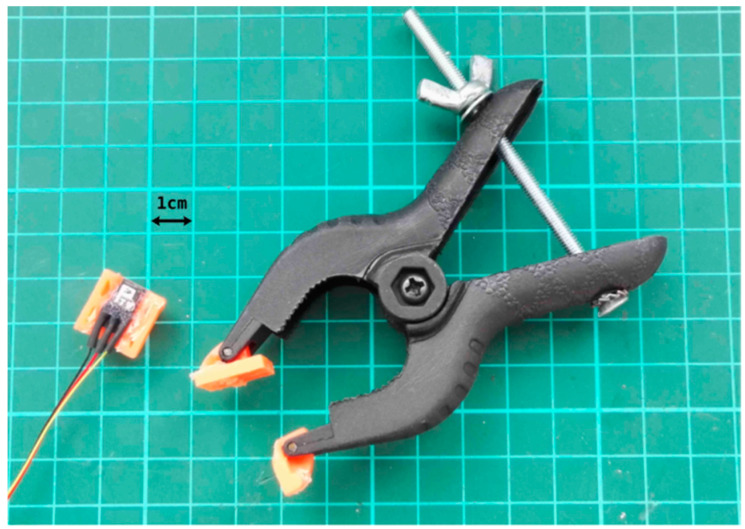
Custom clamp supporting both optical probes. T-probe is shown on the left (cm: centimeter).

**Figure 5 sensors-22-03840-f005:**
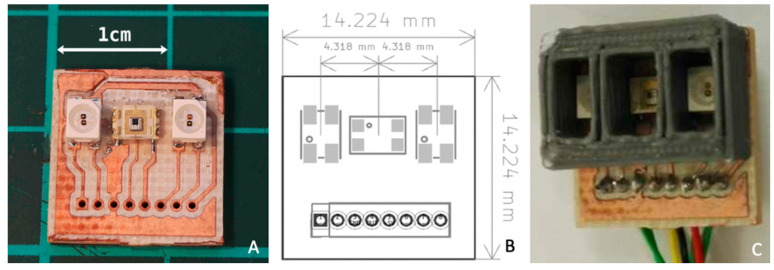
(**A**) Reflectance optode, (**B**) Inter-optode dimensions (**C**) Position of the optical 3 mm barrier (cm: centimeter; mm: millimeter).

**Figure 6 sensors-22-03840-f006:**
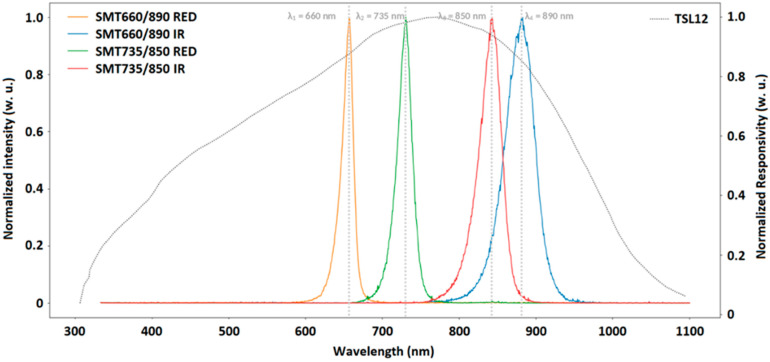
Spectral responses of the 4 LEDs and the photodetector (w. u. means without unit; nm: nanometer).

**Figure 7 sensors-22-03840-f007:**
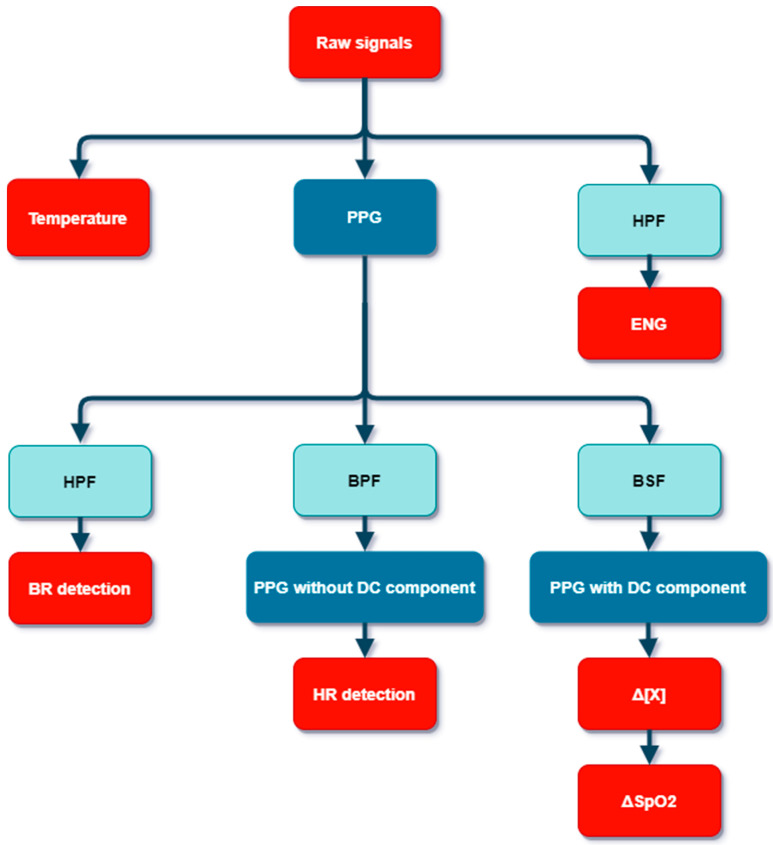
Flowchart of the signal processing (PPG: PhotoPlethysmoGrams; HPF: High-Pass Filter; ENG: ElectroNeuroGraphy; BPF: Band-Pass Filter; BSF: Band Stop Filter; BR: Breath Rate; HR: Heart Rate; DC: Direct Current).

**Figure 8 sensors-22-03840-f008:**
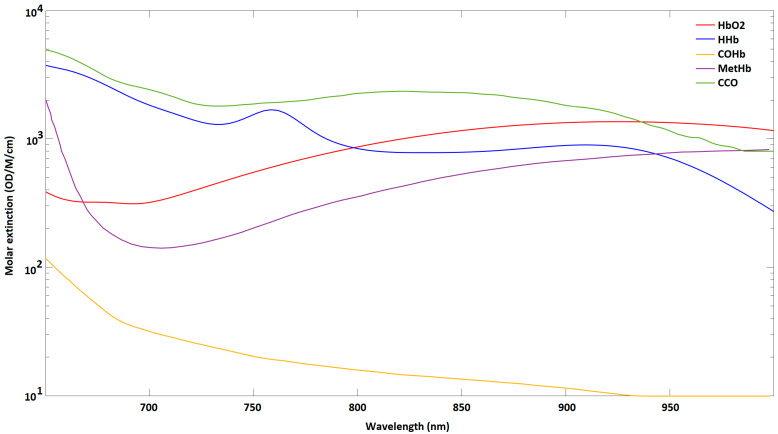
Molar extinction coefficient of five different blood chromophores in function of the light wavelength based on the literature [21] (OD: Optical Density; M: mol/L; cm: centimeter).

**Figure 9 sensors-22-03840-f009:**
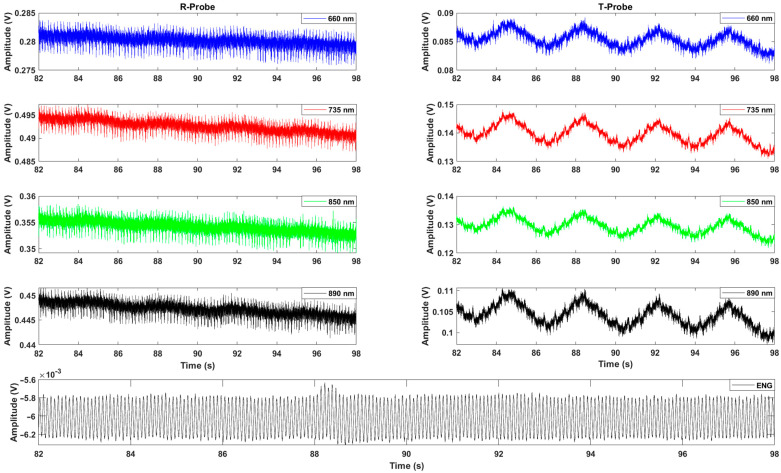
Temporal evolution of the raw PPG signals in baseline, obtained simultaneously with the reflection mode (R-probe) and the transmission mode (T-probe), for the four selected wavelengths and raw biopotential (s: second; V: volt).

**Figure 10 sensors-22-03840-f010:**
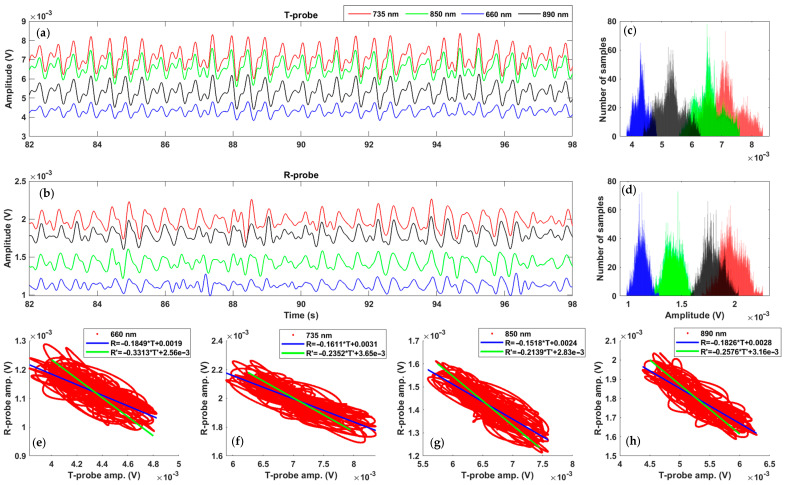
PPG results in baseline for the four wavelengths in both modes ((**a**): transmission, (**b**): reflection) and example of signal processing result ((**c**,**d**): histogram of both PPG, (**e**–**h**): PPG clouds of dots allowing the calculation of the Bravais–Pearson correlation coefficients and regression analysis).

**Figure 11 sensors-22-03840-f011:**
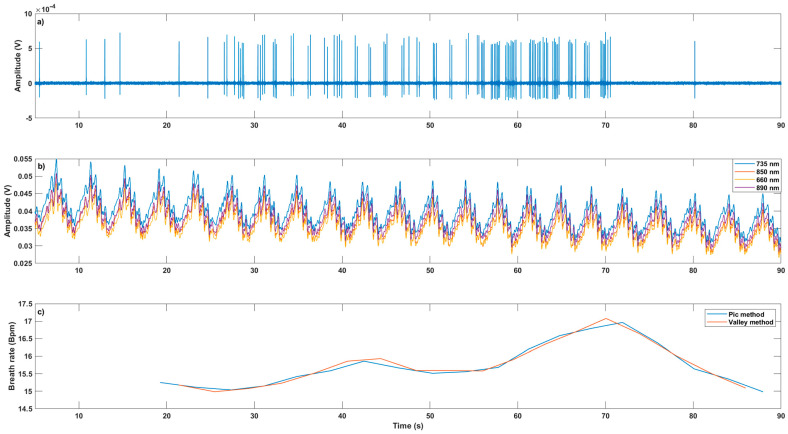
Temporal evolution of optical responses in transmission mode (**b**) and breath rate (**c**) at the spinal cord level with the electrostimulation (**a**) (Bpm: Breaths per minute).

**Figure 12 sensors-22-03840-f012:**
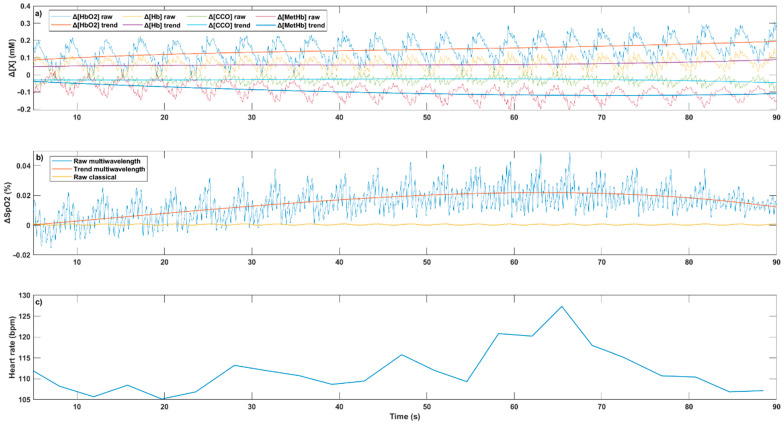
Variation in the chromophore's concentration (**a**), oxygen level (**b**) and heart rate (**c**) at the spinal cord level with the electrostimulation (mM: mmol/L; %: percent; bpm: beats per minute).

**Figure 13 sensors-22-03840-f013:**
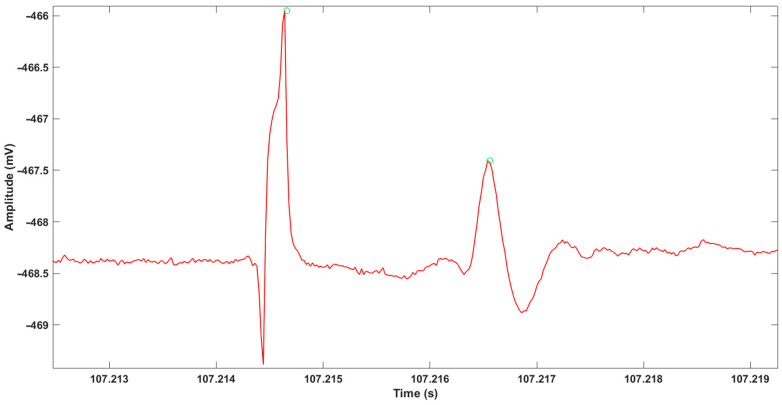
Variation in biopotential at the level of the spinal cord (mV: millivolt; s: second).

**Table 1 sensors-22-03840-t001:** Molar extinction coefficient extracted from Figure 8. Molar extinction coefficient is expressed in OD/M/cm (HbO_2_: oxygenated hemoglobin, HHb: deoxygenated hemoglobin, MetHb: methemoglobin, CCO: cytochrome aa3).

Wavelength	$\varepsilon_{HbO_{2}}$	$\varepsilon_{HHb}$	$\varepsilon_{MetHb}$	$\varepsilon_{CCO}$
λ_1_: 735 nm	464.5	1295.6	168.8	1800.5
λ_2_: 850 nm	1159.3	785.9	530.9	2289.9
λ_3_: 660 nm	334.5	3439.9	609.4	4399.8
λ_4_: 890 nm	1313.4	866.8	890.9	1958.9

**Table 2 sensors-22-03840-t002:** Differential Pathlength Factor in both modes (number without unit).

Wavelength	Reflection	Transmission
λ_1_: 735 nm	6.21	3.49
λ_2_: 850 nm	6.08	3.41
λ_3_: 660 nm	5.00	2.80
λ_4_: 890 nm	3.56	2.00

**Table 3 sensors-22-03840-t003:** Correlation analysis of reflection mode versus transmission mode. (w. u.: without unit; °: degree).

Wavelength	Bravais–Pearson Coefficient (w. u.)	Angle (°)
λ_1_: 735 nm	−0.75	4.1
λ_2_: 850 nm	−0.83	3.4
λ_3_: 660 nm	−0.84	7.9
λ_4_: 890 nm	−0.84	4.1

## Data Availability

The datasets analyzed during the current study are available from the corresponding author on reasonable request.

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
