# Peer review of "Intraoperative Optical Monitoring of Spinal Cord Hemodynamics Using Multiwavelength Imaging System"

_sensors, 2022, doi:10.3390/s22103840_

Round 1
Reviewer 1 Report
This paper has some minor problems and I recommend the following point need to be clarified in the manuscript.
- Where does the data in Figure 8 come from?
- What is the effect of the overall downward tilt of the spectrum in Figure 9? What does the bottom figure of Figure 9 represent?
- The format of the article needs to be revised. For example, for the line break between Line202-203, “figure 13” in Line 361, the last sentence of Conclusions is missing a full stop.
Author Response
We would like to thank the reviewers for their comments.
We hope to have answered them as accurately as possible in order to make our article ready for publication.
Moreover, we did several rounds of proofreading by the co-authors to improve the article.
Response to the reviewer’s comments Number 1
Comments and Suggestions for Authors
This paper has some minor problems and I recommend the following point need to be clarified in the manuscript.
1-Where does the data in Figure 8 come from?
Response to the reviewer's comment:
You are right, we forgot to specify where we got these values from. They are from the media part 1 of the reference “Kolyva, C., Tachtsidis, I., Ghosh, A., Moroz, T., Cooper, C. E., Smith, M., & Elwell, C. E. (2012). Systematic investigation of changes in oxidized cerebral cytochrome c oxidase concentration during frontal lobe activation in healthy adults. Biomedical optics express, 3(10), 2550–2566. https://doi.org/10.1364/BOE.3.002550 ».
We have integrated this reference in the text and in the bibliography.
We have been careful not to make mistakes when integrating it in the bibliography.
2-What is the effect of the overall downward tilt of the spectrum in Figure 9? What does the bottom figure of Figure 9 represent?
Response to the reviewer's comment:
We have specified in the text what this decrease observed in figure 9 corresponds to: “As can be seen, these raw in-vivo signals are heavily noisy, but a trend is emerging for PPGs with periodic pattern and a slight decrease in the mean value of the signals, typically a sign of oxygen desaturation in this transient phase”. It is a period of oxygen desaturation. The curves alternate between phases of decrease and phases of growth according to the global oxygenation rate. The bottom of figure 9 correspond to raw ElectroNeuroGram (ENG).
3-The format of the article needs to be revised. For example, for the line break between Line 202-203, “figure 13” in Line 361, the last sentence of Conclusions is missing a full stop.
Response to the reviewer's comment:
We have taken your comments into account. We have also revised the whole article as requested and made many corrections on the format of the article.

Reviewer 2 Report
Authors have developed a sensor to monitor spinal cords hemodynamics, evaluating two spectroscopic alternatives based on Near Infrared Spectroscopy: Reflectance and Transmittance. The authors have described all the steps carried out in the development of the device including electronics and chemometrics. In addition, after that have tested with pigs the developed sensor with some initial results promising. In my opinion this paper could be published after minor revisions, because there are some items that authors should explain:
- When working with Near Infrared Devices for quantitative or qualitative analysis, instrumentation measures the ratio Intensity of samples/intensity reference material. Do the authors use any internal reference to collect sample spectra? Why do you consider that is it not necessary in your research work? Please explain this details in the text.
- Authors show in Table 1 the values of the Molar Extinction Coefficient from Figura 8, however those values are difficult to understand. The ordinate axis (fig. 8) shows values of Molar extinction coefficient ranged between 10 to 104 . However in Table 1 those values are ranged between 0.01 to 3.44, and all values should be higher than 10. Perhaps it is due to the measurements unit which are not showed in Table 1. Please revise these data.
- For better understanding in Tables 2 and 3 include Wavelength values.
- In line 280 authors state that: “the reflection measurements suffered from saturation on certain wavelengths (not presented here)…”. Please dtail in the text which wavelengths suffer saturation and the reasons of that saturation.
- Legend Figure 12: Authors do not explain the information showed in Figure 12. There are three Figures, a; b and c, but in the legend it is not detailed.
Author Response
We would like to thank the reviewers for their comments.
We hope to have answered them as accurately as possible in order to make our article ready for publication.
Moreover, we did several rounds of proofreading by the co-authors to improve the article.
Response to the reviewer’s comments Number 2
Comments and Suggestions for Authors
Authors have developed a sensor to monitor spinal cords hemodynamics, evaluating two spectroscopic alternatives based on Near Infrared Spectroscopy: Reflectance and Transmittance. The authors have described all the steps carried out in the development of the device including electronics and chemometrics. In addition, after that have tested with pigs the developed sensor with some initial results promising. In my opinion this paper could be published after minor revisions, because there are some items that authors should explain:
1-When working with Near Infrared Devices for quantitative or qualitative analysis, instrumentation measures the ratio Intensity of samples/intensity reference material. Do the authors use any internal reference to collect sample spectra? Why do you consider that is it not necessary in your research work? Please explain this detail in the text.
Response to the reviewer's comment:
In our preliminary work, elementary materials were used and preliminary tests on reference elements were indeed carried out. Optical spectroscopy measurements have even been performed during in-vivo test ([16] Tsiakaka, O.; Feruglio, S. Toward the Monitoring of the Spinal Cord: A Feasibility Study. Microelectronics Journal, 2019, 88, 145–153. https ://doi.org/10.1016/j.mejo.2018.01.026.) and optical elements have also been characterized ([19] Tsiakaka, O.; Gosselin, B.; Feruglio, S. Source-Detector Spectral Pairing-Related Inaccuracies in Pulse Oximetry: Evaluation of the Wavelength Shift. Sensors (Basel), 2020, 20 (11), E3302. https://doi.org/10.3390/s20113302.)
We are now working in relative value, thanks to equation (2) to minimize calibration during in-vivo experiments. This is why we provide readers with variations in Figure 12.
2-Authors show in Table 1 the values of the Molar Extinction Coefficient from Figure 8, however those values are difficult to understand. The ordinate axis (fig. 8) shows values of Molar extinction coefficient ranged between 10 to 104. However, in Table 1 those values are ranged between 0.01 to 3.44, and all values should be higher than 10. Perhaps it is due to the measurements unit which are not showed in Table 1. Please revise these data.
Response to the reviewer's comment:
Indeed, presented like this it is difficult to make the link between table 1 and figure 8. It is, as you have envisaged, a difference of unit. We have therefore reproduced Table 1 using the same unit for greater clarity.
In addition, when we went back to the table we realized that there was an error in the values in the last column.
The corrections were made as requested.
3-For better understanding in Tables 2 and 3 include Wavelength values.
Response to the reviewer's comment:
Thanks for your comment, we have improved the accuracy of the tables by adding the wavelengths as requested.
4-In line 280 authors state that: “the reflection measurements suffered from saturation on certain wavelengths (not presented here) …”. Please detail in the text which wavelengths suffer saturation and the reasons of that saturation.
Response to the reviewer's comment:
This is an interpretation error due to a lack of precision in the text. When we performed the measurements in reflexion, many of the measurements suffered from saturation due to bleeding from this type of surgical procedure.
We have changed the terms to improve the understanding.
5-Legend Figure 12: Authors do not explain the information showed in Figure 12. There are three Figures, a; b and c, but in the legend, it is not detailed.
Response to the reviewer's comment:
Indeed, the legend of figure 12 was not very precise. We have made the corrections as requested.
